# MicroRNAs: Potential Biomarkers of Disease Severity in Chronic Rhinosinusitis with Nasal Polyps

**DOI:** 10.3390/medicina59030550

**Published:** 2023-03-11

**Authors:** Anda Gata, Ioana Berindan Neagoe, Daniel-Corneliu Leucuta, Liviuta Budisan, Lajos Raduly, Veronica Elena Trombitas, Silviu Albu

**Affiliations:** 1Department of Otorhinolaryngology, University of Medicine and Pharmacy “Iuliu Hatieganu”, 400349 Cluj-Napoca, Romania; 2Research Center for Functional Genomics, Biomedicine and Translational Medicine, University of Medicine and Pharmacy “Iuliu Hatieganu”, 400349 Cluj-Napoca, Romania; 3Medical informatics and Biostatistics Department, University of Medicine and Pharmacy “Iuliu Hatieganu”, 400349 Cluj-Napoca, Romania

**Keywords:** chronic rhinosinusitis, nasal polyp, microRNA-125b, microRNA-203a-3p, biomarker, endotype

## Abstract

*Background and Objectives*: Chronic rhinosinusitis with nasal polyps (CRwNP) has multiple clinical presentations, and predictors of successful treatment are correlated to different parameters. Differentially expressed microRNAs in nasal polyps emerge as possible facilitators of precise endotyping in this disease. We aimed to evaluate the correlation between the clinical parameters of CRSwNP and two different microRNAs. *Materials and Methods*: The expression of miR-125b and miR-203a-3p in nasal polyps (*n* = 86) and normal nasal mucosa (*n* = 20) was determined through microarray analysis. Preoperative workup included CT scan, nasal endoscopy, blood tests, symptoms and depression questionnaires. *Results*: MiR-125b showed significant overexpression in NP compared to the normal nasal mucosa. miR-125b expression levels were positively and significantly correlated with blood eosinophilia (*p* = 0.018) and nasal endoscopy score (*p* = 0.021). Although high CT scores were related to miR-125b overexpression, the correlation did not reach statistical significance. miR-203a-3p was underexpressed in nasal polyps and was significantly underexpressed in CRSwNP patients with environmental allergies. *Conclusions*: Both miR-125b and miR-203a-3p are potential biomarkers in CRSwNP. miR-125b also correlates with the clinical picture, while miR-203a-3p could help identify an associated allergy.

## 1. Introduction

Chronic rhinosinusitis (CRS) is defined by the European Position Paper on Rhinosinusitis and Nasal Polyps (EPOS20) guideline as mucosal inflammation with a duration of more than 12 weeks, and it is considered a significant health issue worldwide [1]. CRS represents a burden for patients, with significant impact on quality of life, and it has an incidence of 10.9% in European countries [2]. Currently, the phenotypic classification divides CRS according to the presence (CRSwNP) or absence (CRSsNP) of nasal polyps. Moreover, according to the percentage of eosinophils present in nasal polyps, CRSwNP can be further classified as eosinophilic (Eos) or non-eosinophilic (non-Eos) [1,3]. Traditionally, CRSsNP is associated with fibrosis and T helper type 1 (Th1) inflammation, which is characterized by the presence of neutrophils, type 1 cytokines such as interleukin-2 (IL-2), and interferon-γ (IFN-γ), whereas Th2 inflammation, with eosinophils (Eos) and type 2 cytokines, such as IL-4, IL-5, and IL-13, is present in CRSwNP [4]. However, the delineation between type 1 and 2 is not precise, and CRSwNP is a heterogenous disease with various clinical presentations and high ethnic variability. In Caucasians, there is a predominantly T2 response characterized by edema, while Asian subjects have a preponderant T1/T17 pattern of inflammation [1,3,5,6,7]. In the last decades, many studies have attempted to classify CRSwNP further according to endotypes that could guide future treatment strategies.

The pathogenesis of CRS was thought to be influenced by genetic and environmental factors [8]. The field of epigenetics has contributed greatly to the future of precision medicine and the discovery of microRNAs (miRNAs). MiRNAs are a family of small, non-coding RNAs which function as post-transcriptional regulators by binding to mRNA. Consequently, this leads to cleavage and the induction of degradation or blockage of translation of target mRNA, participating in a wide range of biological processes, such as development, stress response, signal transduction, apoptosis and cancer. They are being considered the main regulatory mechanisms of gene expression [9,10]. They play a significant role in cell division, differentiation and development, directly modulating innate immune responses [9]. The modulation of inflammatory response by miRNAs has been demonstrated in multiple sclerosis, psoriasis, inflammatory bowel disease, rheumatoid arthritis, and different cancers [11]. The potential implications of miRNAs in chronic rhinosinusitis have only recently emerged, and they have become the main research focus worldwide.

In CRSwNP, miRNA’s aberrant expression is associated with activating pro-inflammatory pathways, leading to inflammation and fibrosis, with the subsequent formation of nasal polyps [12,13]. Multiple studies have discovered an overexpression of miR-125b in nasal polyps [12,14] and even a distinct expression between Eos and non-Eos CRSwNP [15]. Since miR-125b appears to be one of the most mentioned microRNAs in CRSwNP, with expression in NP compared to normal nasal mucosa, and even higher values in EosCRSwNP, we find it useful to also evaluate the correlation between CRSwNP clinical severity and miR125b values. Moreover, increased serum levels of miR-125b were also found in asthma patients [10], along with miR-203a-3p, which proved to have potential involvement in the epithelial–mesenchymal transition (EMT) process related to the etiopathology of asthma [16]. Having the unified airway concept in mind, the aim of this study was to determine the expression of miR-125b and miR203a-3p in nasal polyps, compared with normal nasal mucosa, and correlate the miRNA expression with disease aggressiveness, finally aiming to investigate their possible role as biomarkers for CRSwNP. Since miR-203a-3p is a proven participant in EMT but has not yet been investigated in nasal polyps, we considered investigating its potential as a biomarker in CRSwNP. In the future, the determination of multiple miRNA, as biomarkers, could help the clinicians delineate the particularities of each CRSwNP patient and facilitate tailored treatments for this heterogenous pathology. In addition, miRNAs could also become therapeutic targets in the course of time.

## 2. Materials and Methods

### 2.1. Study Design and Setting

This study was conducted in accordance with the principles of the Declaration of Helsinki and was approved by our Institutional Ethics Committee (590/10 December 2019). Patients with CRSwNP undergoing endoscopic sinus surgery at the tertiary hospital CF Cluj Clinical Hospital, University of Medicine and Pharmacy ‘Iuliu Hațieganu’ Cluj Napoca, Romania, were prospectively and consecutively recruited between January 2020 and November 2021.

### 2.2. Participants

A total of total of 106 patients were prospectively included in this study: 86 patients with CRSwNP (29 women, 57 men) and 20 control patients (8 women, 12 men) without CRS. Diagnosis of CRSwNP was obtained according to the European Position Paper on Rhinosinusitis and Nasal Polyps criteria EPOS2012 [17]. In our department, according to EPOS2012, only patients failing maximal medical treatment are proposed for surgical intervention. In addition, only patients with CRSwNP proposed for primary surgery were included; all patients with revision surgery were excluded. Patients with antro-choanal polyps, unilateral sinusitis, cystic fibrosis, primary ciliary dyskinesia, immunodeficiencies and tumor suspicion were excluded. In addition, patients using systemic corticosteroids or macrolides in the previous month or concurrent immunomodulatory therapy were excluded. All patients signed informed consent before enrollment.

### 2.3. Variables and Measurements

Preoperative data collection included demographic data (age, gender), smoking history, concomitant asthma, environmental allergies, and hypersensitivity to nonsteroidal anti-inflammatory drugs (NSAIDs), nasal polyp endoscopic grading using Lildholdt scale [18]; extension on computed tomography was assessed using Lund–Mackay score [19], and high blood eosinophilia was defined as values over 5% of total white blood cells. All patients filled in preoperative SNOT-22 and PHQ-9 depression questionnaires.

The decision to undergo surgery and all clinical investigations were made following the disease management, and they were not part of this study’s protocol. Nasal polyp tissue from the middle meatus was obtained from CRSwNP patients; inferior turbinate mucosa was used as control and obtained from patients undergoing surgical treatment for septal deviation without associated inflammatory nasal disease such as rhinitis or sinusitis. The samples were immediately frozen in liquid nitrogen and transported to the Genomics Medical Center for miRNAs’ analysis.

### 2.4. MiRNAs’ Analysis

#### 2.4.1. RNA Extraction

Total RNA was extracted and isolated from tissue samples using the phenol-chloroform (Trireagent) method according to the manufacturer’s protocol. The concentration of RNA was determined by NanoDrop-1000 spectrophotometer and ranged to 50 ng/μL.

#### 2.4.2. cDNA Synthesis and qRT-PCR

The cDNA synthesis was performed using 50 ng of total RNA according to TaqMan MicroRNA Reverse Transcription Kit (Applied Biosystems) protocol. The cDNA mixture was incubated in PCR tubes at 16 °C 30 min, 42 °C 30 min, and 85 °C 5 min.

qRT-PCR was performed in a total volume of 10 μL of cDNA using TaqMan Fast Advanced Master MIX (Applied Biosystems) according to the manufacturer protocol and 0.73 μL primer for each miRNA in ViiA7 (Applied Biosystems) PCR machine. The miRNA primer sequences of qRT-PCR are listed in Table 1.

The relative expression level was calculated using −ΔΔCT and fold change 2^−ΔΔCT^, and for expression normalization, U6 and RNU48 were used.

### 2.5. Statistical Analysis

Categorical data were presented as absolute and relative frequencies. Quantitative data were presented as means and standard deviations, or medians, and 25 and 75 percentiles in case of a normal distribution or non-normal distribution. Comparisons between two independent groups concerning qualitative data were performed with the chi-squared test or Fisher exact test (in case of low expected frequencies). Comparisons between two independent groups concerning quantitative data were performed with a t-test for independent samples (in case of normal distribution) or a Wilcoxon rank-sum test (in case of non-normal distribution). To assess the classifying ability of miR-125 and miR-203, receiver operating characteristic curves were plotted, and the area under the curve with bootstrapped 95% confidence intervals was computed, while the comparison between the two curves was carried out with a DeLong test. For miR-125 and miR-203, we used their medians to transform them into binary variables, indicating overexpression or underexpression. Correlations between quantitative variables were assessed with the Spearman correlation coefficient, bootstrapped confidence intervals and the associated statistical test. All statistical analyses were carried out in R environment for statistical computing and graphics (R Foundation for Statistical Computing, Vienna, Austria), version 4.1.2 (R Core Team. R: A Language and Environment for Statistical Computing. R Foundation for Statistical Computing: Vienna, Austria; 2022).

## 3. Results

A total of 89 patients with CRSwNP and 20 control subjects were enrolled. After tissue processing for miRNA assay, three CRSwNP samples were inadequate for examination, and we excluded the patients. Demographic data are presented in Table 2. In the CRSwNP group, 28 patients were associated with asthma, 14 of which also presented hypersensitivity to NSAIDs. Environmental allergies were positive for 18 patients, and 37 presented high peripheral blood eosinophilia. miR-203a-3p was underexpressed in CRSwNP patients with associated environmental allergies (Table 3).

After qRT-PCR analysis, the results showed significantly higher levels of miR-125b (Figure 1a) and significant underexpression of miR-203a-3p (Figure 1b) in nasal polyps compared to normal mucosa.

To verify the diagnostic ability of the two miRNAs in classifying polyposis vs. normal subjects, we computed and plotted an operator characteristic curve (Figure 2). miR-203a-3p had better classification accuracy compared to miR-125b, albeit not statistically significant (*p* = 0.146).

Statistically significant overexpression of miR-125b was found in patients with higher nasal polyp endoscopy scores and high blood eosinophil levels. Elevated miR-125b values were also correlated with high Lund–Mackay scores but without reaching statistical significance. SNOT-22 values do not correlate with miR values, as shown in Table 4.

The group of CRSwNP without asthma presented lower values of miR-125b compared with patients associating asthma. There was a difference of 0.17 (95% CI −0.58–0.19), *p* = 0.478 (Mann–Whitney U test) between mean values, not reaching statistical significance (Figure 3a).

While testing the global difference between patients with CRSwNP and associated environmental allergy or NSAIDS hypersensitivity, miR-125b showed slightly higher values for patients with NSAID hypersensitivity but did not reach statistical significance (Figure 3b).

A value of 0.37 (95% CI 0.02–0.56) Spearman correlation coefficient was found between the percentage of peripheral blood eosinophils and miR-125b in CRSwNP patients, results being statistically significant, *p* = 0.028 (Figure 4).

## 4. Discussion

Our prospective cohort study found statistically significant differences between polyposis and normal subjects concerning miR-125 and miR-203; miR-203 had slightly better classifying abilities than miR-123. miRNA-203 was underexpressed in subjects associated with environmental allergies. The overexpression of miR-125 was associated with endoscopy scores and eosinophils.

Chronic rhinosinusitis with nasal polyps results from complex inflammatory pathways and diverse genetic components, leading to numerous clinical presentations. In the past decades, there has been a serios focus on delineating different endotypes of this pathology, finding biomarkers that could predict disease aggressiveness, and tailoring the treatment for best disease control. Recently, there has been significant effort in identifying the precise mechanisms leading to polyp formation, and miRNAs have emerged as critical performers.

MiR-125b is one of the most mentioned non-coding RNA in CRS. Zhang et al. [15] were the first to investigate miR-125b expression in patients with chronic rhinosinusitis nasal tissues. The study included 170 patients, which were distributed in groups of chronic rhinosinusitis without nasal polyps (CRSsNP), Eos CRSwNP, non-Eos CRSwNP and normal nasal mucosa. The significant overexpression of miR-125b was observed only in Eos CRSwNP but not in other types of CRS nor was miR-125b differentially expressed in non-Eos CRS compared to control subjects. These results do not agree with other studies [12,14], including ours, where a clear overexpression of miR-125b was observed in nasal polyps compared to the normal nasal mucosa. When comparing the expression of seven different miRNAs, Xia et al. [14] found miR-125b overexpressed in CRS compared to controls and further elevated in CRSwNP compared to CRSsNP. Song et al. [12] performed a microarray analysis, screening for differentially expressed miRNAs in nasal tissues of CRSwNP compared to CRSsNP, and found miR-125b to have the highest change in NP and also be correlated with eosinophilic infiltration. During a general analysis of microRNAs in CRSwNP, Yu et al. [13] found differentiated miRNAs expression in CRS compared to control but without difference according to tissue eosinophilia. Mucosal eosinophilic status is in direct correlation with the severity and outcome of CRSwNP. It is worldwide accepted that tissue Eos CRSwNP can be diagnosed through tissue histopathology eosinophil count or quantification of eosinophil-derived mediators (e.g., eosinophil cationic protein (ECP)). Yet, consensus regarding the cutoff value defining tissue eosinophilia has not been reached, the method can be considered subjective, and not all centers have the necessary infrastructure. A viable alternative for diagnosing Eos CRSwNP seems to be blood eosinophilia, since there has been clear proof that high blood Eos correlates with nasal polyp infiltrating eosinophil counts [20,21,22,23]. As a noninvasive parameter, a Lund–Mackay scoring system on CT scan was presented as a predictor of Eos-CRSwNP with a specificity of 90% and a sensitivity of 94%. (Meng). A multi-institutional study from Japan (JESREC) [6] created an algorithm to classify CRS into non-EosCRS, mild, moderate and severe EosCRS, based on blood eosinophilia and extent of disease on computed tomography. In addition, this classification correlated with the rate of recurrence and refractoriness. In our study population, eosinophilia was diagnosed based on peripheral blood Eos values, and there were significantly higher values of miR125-b in patients with nasal polyps and high Eos count. Moreover, an overexpression of miR-125b also correlated with high endoscopic scores of nasal polyps, and high Lund–Mackay scores, however not reaching statistical significance for the latter. Interestingly, we could not observe any direct relation between symptom scores (SNOT-22) and miR-125b values. Anyhow, it has already been demonstrated that patient symptom scores do not necessarily correlate with objective evaluations of disease severity (peak nasal inspiratory flow rates, CT scan scores [24]. Patient symptom scores are subjective measurements that are also influenced by other factors, such as the mental well-being of patients; thus they could not precisely correlate with disease severity. Having this in mind, we could interpret that miR-12b and miR-203a-3p did not correlate with symptom scores due to the subjectiveness of the latter. Furthermore, in a preliminary article based on the same patient population, we showed that SNOT-22 was also influenced by depression [25]. Significantly higher symptom scores were correlated with elevated scores on PHQ-9 depression questionnaire and did not reflect the CRSwNP disease severity.

Sensitivity, specificity, positive and negative predictive values are all desirable performance traits of a biomarker [26]. Additionally, it should be somewhat safe, cost-effective, and simple to measure. In certain ways, miRNAs appear to be a potential area for research in this domain. Panganiban et al. [27] managed to correctly differentiate with a sensitivity of 92.4%, between healthy, asthmatic or allergic rhinitis subjects through blood microarray analysis. The results were obtained by a random forest prediction model based on the six most relevant microRNAs (including miR-125b) modified in these diseases. In our study, miR-125b expression was slightly upregulated in patients associating asthma but without statistical significance. We found three explanations for this discordance: one would be that miR-125b was determined from nasal polyps, while other studies, which showed its overexpression in asthma, used serum levels or tracheal biopsies [27,28,29]. Second, there might overlap between expressions, since both CRSwNP and asthma, taken separately, show modified levels of miR125b. Lastly, the number of patients with concomitant asthma included in the study might need to be increased to reach statistical significance.

Epithelial-to-mesenchymal transition (EMT) is a cellular process recognized as a key element in tissue remodeling [30]. Transforming growth factor- β1(TGF-β1) induces EMT in lower airways via Smad3, leading to asthma or chronic obstructive pulmonary disease [31]. In nasal epithelial cells, TGF-β1 is claimed to promote EMT via the upregulation of α-smooth muscle actin, leading to polyp formation. [32] Furthermore, Fan et al. [16] revealed that miR-203a-3p is underexpressed and regulates TGF-β1-induced EMT in asthma patients. By targeting SIX1, miR-203a-3p, regulates the Smad3 pathway, thus inhibiting TGF-β1 and stopping the induction of EMT in bronchial epithelial cells. Even though these experiments were only conducted in vitro, miR-203a-3p is a promising therapeutic target. Tsai et al. also observed the downregulation of this microRNA [33] in asthma patients using next-generation sequencing. Having this common pathway in mind, we subsequently measured miR-203a-3p in nasal polyps and found a significant underexpression compared to the normal nasal mucosa. We could not correlate the results with the presence of concomitant asthma, but we found an expression influenced by coexisting environmental allergies. A literature review investigating the relationship between allergy and chronic rhinosinusitis found a poor correlation between the two entities. However, there is a strong association between allergy and certain CRSwNP subtypes, such as allergic fungal rhinosinusitis and central compartment atopic disease [34]. It is possible that with better defined patient categories, miR-203a-3p could be a tool for predicting the atopic status of CRSwNP patients. On the other hand, a recently published study from Poland [35] described higher values of miR-203a-3p in CRSwNP patients, compared to controls, yet the RT-qPCR analysis was performed on the maxillary sinus mucosa of 10 CRSwNP patients and not on nasal polyps. Additional research would be warranted for clarifying the association between miR-203a-3p and nasal polyposis.

In addition to the potential role as biomarkers, microRNAs in CRS could be potential therapeutic targets in the future, but their exact mechanism of action must be better understood. To date, Zhang et al. [15] revealed a potential role of miR-125b in airway antiviral innate immunity. The MiR-125b-4E-BP1 pathway may contribute to mucosal eosinophilia in CRS by promoting type I IFN expression. They demonstrated that 4E-BP1, a translational repressor, is a direct target of miR-125b, and 4E-BP1 protein could be revealed by nasal epithelial cells. An interesting study [36] validated 4E-BP1 as a key component for innate immune activation. They have shown that 4E-BP1 represents a negative regulator of virus-evoked type-I IFN production through the translational repression of IRF-7 mRNA. Zhang et al. [15] confirmed these findings and demonstrated that miR-125b antagomir treatment increases the expression of 4E-BP1 protein and inhibits INF-β expression in murine nasal cells. They discovered a potential link between eosinophilic inflammation in sinonasal mucosa, IFN-β mRNA expression, and IL-5 mRNA, pointing to the possibility of a connection between type I IFN and Th2-biased eosinophilic inflammation in sinonasal mucosa. Interestingly, IFN-β treatment has been shown to shift the pattern of the immune response from Th1 to Th2 in multiple sclerosis [37,38]. Song et al. [12] also confirmed that miR-125b was in direct correlation with eosinophil cationic protein (ECP) and IL-5, IL-8 levels in CRSwNP patients. Moreover, they demonstrated in murine tissues that a downregulation of miR-125b leads to a reduction in ECP, eosinophil peroxidase (EPX), IL-5 and IL-8 levels.

As far as the molecular mechanisms, it is well known that intense edema with pseudocyst formation plays a major role in NP formation. Inflamed tissues exhibit higher vessel permeability, which leads to plasma exudation, which is then retained by the fibrin deposits also generated by inflammation, hence forming the typical NP pseudocysts. [39] Exaggerated fibrin deposition in NP appears to be a consequence of abberant fibrin degradation. t-PA and u-PA divide plasminogen, leading to plasmin formation, which in turn is the key facilitator of fibrin degradation [12]. Song et al. [12] demonstrated in a murine model that the Sp1/miR-125b axis is responsible for generating inflammation and fibrin deposition in NPs, with the involvement of the Wnt/β-catenin signaling pathway. They witnessed that the downregulation of miR-125b increased the levels of d-dimer and the level of t-PA, concluding that miR-125b might lead to fibrin deposition in NPs. Sp1 is a transcription factor with various implications in disease pathophysiological mechanisms and is a key regulator of fibrosis through the regulation of TGF-β1/Smad pathways. Increased levels of Sp1 were also observed in nasal polyps. They described a reduction in d-dimers and t-PA levels when Sp1 was upregulated. In addition, miR-125b inhibition reversed all these processes. Wnt/β-catenin is the other pathway discovered by Song et al. to be in direct relation with miR-125b. This pathway is directly related to inflammation, the activation of myofibroblasts and consequent tissue fibrosis. [40] Sp1 upregulation increased activation of the Wnt/β-catenin pathway in murine NP tissues and was reduced after miR-125b downregulation. Subsequently, miR-125b appears as a key player in two major molecular pathways implicated in NP formation. However, additional experimental research is required to determine whether and how these miRNAs are involved in the regulation of CRS.

We acknowledge that our study’s major limitation is the need for more tissue eosinophil analysis, which would have given more precise delineation between Eo and non-Eo patients. Furthermore, even though it is a prospective study, a larger patient population would have helped obtain statistical significance when comparing patients associating asthma, NSAID hypersensitivity, or having Samter triad.

Our study strength is the prospective and standardized enrollment of a considerable number of patients, consisting of a fairly equal subdistribution according to the presence or absence of bronchial asthma or allergies.

## 5. Conclusions

MiRNAs are surfacing as key players in CRSwNP. MiR-125b materializes into a possible biomarker for CRSwNP, and it shows promise as a future therapeutic target. Considering the controversy around allergy in CRSwNP, miR-203a3p could contribute, as a non-invasive biomarker, to further delineating the allergic endotypes.

## Figures and Tables

**Figure 1 medicina-59-00550-f001:**
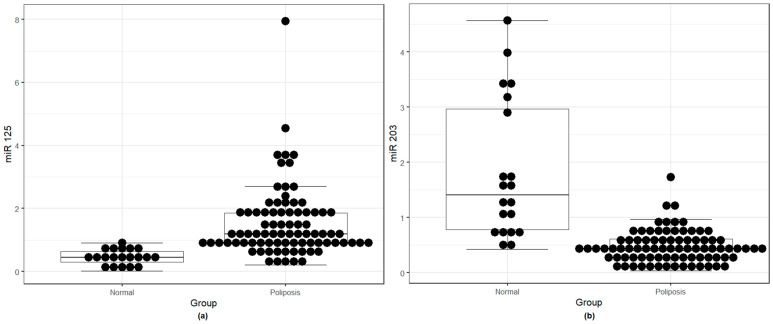
(**a**) miR-125b overexpression in nasal polyps compared to the normal nasal mucosa; (**b**) miR-203a-3p underexpression in nasal polyps compared to the normal nasal mucosa.

**Figure 2 medicina-59-00550-f002:**
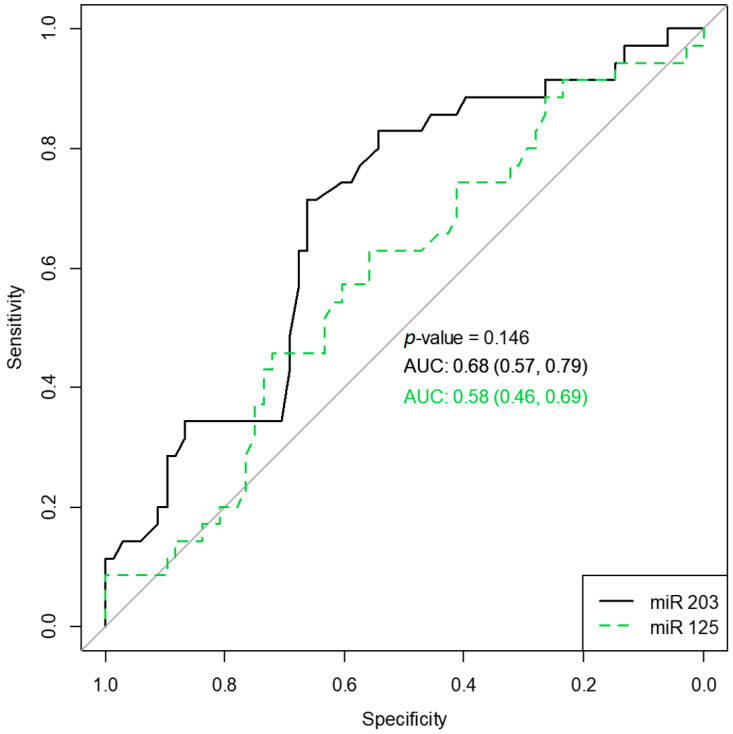
Receiver operating characteristic curves for miR203 and miR125 classifying polyposis vs. normal subjects. AUC, the area under the curve, along with 95% confidence intervals.

**Figure 3 medicina-59-00550-f003:**
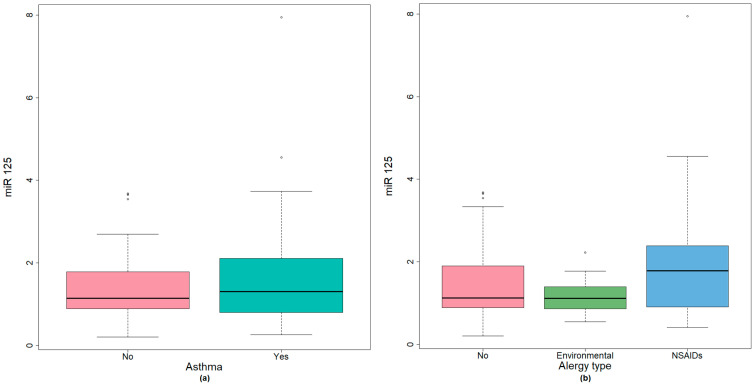
(**a**) miR-125 expression in CRSwNP patients with environmental allergy or NSAIDs hypersensitivity; (**b**) miR-125 expression in CRSwNP with associated bronchial asthma.

**Figure 4 medicina-59-00550-f004:**
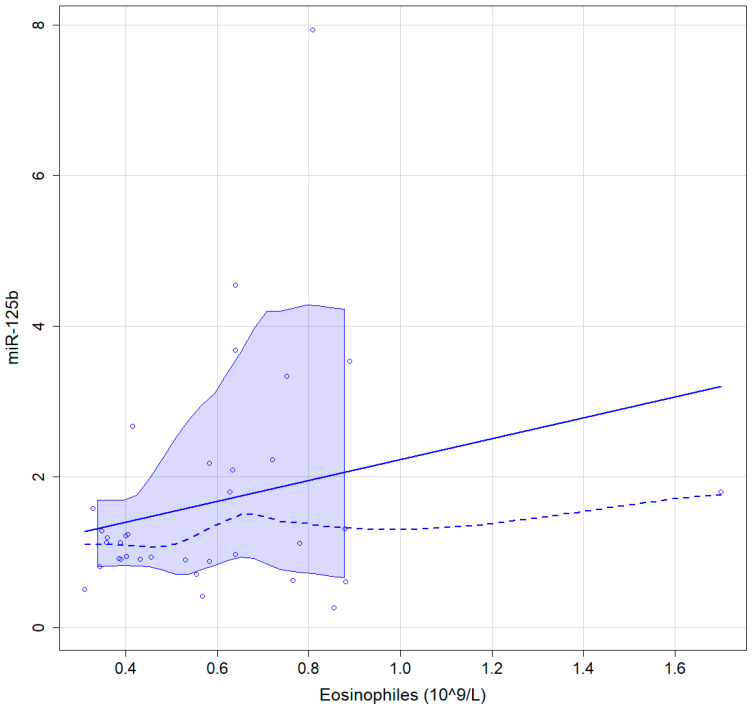
Graphic representation of the association between blood eosinophiles and miR-125b continuous line, linear regressions interrupted line, locally estimated scatterplot smoothing with 95% confidence interval.

**Table 1 medicina-59-00550-t001:** miRNA primers used in the study.

**hsa-miR-203a-3p**	000507	GUGAAAUGUUUAGGACCACU
**hsa-miR-125b**	000449	UCCCUGAGACCCUAACUUGUGA
**U6 snRNA**	001973	GTGCTCGCTTCGGCAGCACATATACTAAAATTGGAACGATACAGAGAAGATTAGCATGGCCCCTGCGCAAGGATGACACGCAAATTCGTGAAGCGTTCCATATTT
**RNU48**	001006	GATGACCCCAGGTAACTCTGAGTGTGTCGCTGATGCCATCACCGCAGCGCTCTGACC

**Table 2 medicina-59-00550-t002:** Comparison between polyposis and healthy participants.

Group	Normal	Polyposis	Difference (95% CI)	*p*-Value
	(*n* = 20)	(*n* = 86)		
Age (years), mean (SD)	37.12 (12.8)	49.57 (12.76)		<0.001
Sex (F), *n* (%)	8 (40)	29 (33.72)		0.596
Smoking, *n* (%)	3 (15)	20 (23.25)		0.554
Asthma, *n* (%)	0 (0)	28 (32.55)		0.010
Allergy, *n* (%)				0.09
Environmental	3 (15)	18 (20.93)		
NSAIDs	0 (0)	14 (16.28)		
No	17 (85)	54 (62.79)		
miR-125, median (IQR)	0.4 (0.3–0.64)	1.19 (0.88–1.85)	0.74 (−1.08–−0.53)	<0.001
miR-203, median (IQR)	1.42 (0.78–2.97)	0.4 (0.26–0.62)	1.02 (0.59–1.37)	<0.001

SD, standard deviation; IQR, interquartile range; CI, confidence interval; NSAIDs, nonsteroidal anti-inflammatory drugs; miR, micro-ribonucleic acid.

**Table 3 medicina-59-00550-t003:** miR-203a-3p underexpressed in subjects with associated environmental allergies.

Environmental Allergy:	No (*n* = 82)	Yes (*n* = 21)	Difference (95% CI)	*p*
miR-125, median (IQR)	0.96 (0.64–1.82)	0.98 (0.71–1.39)	0.03 (−0.24–0.41)	0.658
miR-203, median (IQR)	0.53 (0.34–0.77)	0.3 (0.2–0.52)	0–14 (0–0.31)	0.047

**Table 4 medicina-59-00550-t004:** miR-125b overexpression comparisons in participants with polyposis.

miR-125b Overexpression:	No	Yes	*p*
	(*n* = 41)	(*n* = 45)	
Age (years), median (IQR)	50 (44–61)	46 (36–55)	0.073
Sex (F), nr (%)	18 (43.9)	11 (24.44)	0.057
Smoking (Yes), nr (%)	11 (26.83)	9 (20)	0.454
Allergy (Yes), nr (%)	15 (36.59)	17 (37.78)	0.909
Allergy type, nr (%)			0.187
NSAIDs	4 (9.76)	10 (22.22)	
Environmental	11 (26.83)	7 (15.56)	
No	26 (63.41)	28 (62.22)	
Asthma (Yes), nr (%)	12 (29.27)	16 (35.56)	0.534
Endoscopy score average score, median (IQR)	2 (1.5–2.5)	2.5 (2–3)	0.021
Endoscopy score maxim score, median (IQR)	2 (2–3)	3 (2–3)	0.027
Endoscopy score sum score, median (IQR)	4 (3–5)	5 (4–6)	0.021
Eosinophiles (%), median (IQR)	6.42 (5.62–7.29)	8.37 (6.54–11.09)	0.018
PHQ-9 preop, median (IQR)	4 (2–6)	5 (3–8)	0.107
Lund Mackay sum score, median (IQR)	15 (13–19)	17 (14–20)	0.063
SNOT initial, median (IQR)	36 (26–54)	49 (28–65)	0.093

SD, standard deviation; IQR, interquartile range; CI, confidence interval; NSAIDs, nonsteroidal anti-inflamatory drugs; PHQ-9 Depression test questionnaire; SNOT-22, Sinonasal outcome test; miR, micro-ribonucleic acid.

## Data Availability

No data available due to privacy concerns.

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
