# Peer review of "MicroRNAs: Potential Biomarkers of Disease Severity in Chronic Rhinosinusitis with Nasal Polyps"

_medicina, 2023, doi:10.3390/medicina59030550_

Round 1

Reviewer 1 Report

Dear Authors,

a topical issue for this time. Despite quite nice presentation of issue, there are some points to be improved in this manuscript:
1) Introduction. Please, add the significance of the chosen mRNA, this should be described more in extent way (perhaps you can use some info from the Discussion and move it here);

2) please, put Ethical Committee permission info into the text. Additionally, please move info about the patients age, sex, anamnesis morbi et morborum main points, treatments etc. from Results Line134-140 to the Materials and methods Subject section. They fit here. Additionally, please, describe also the clear inclusion/exclusion criteria here about the patients and controls. Also, add the info were recurrent polyposis case included into the research group (separate then the primary and recurrent polyp group; now it is not understandable).

3) Conclusions. Please, make them more precise, - remove the 1st sentence, what is extra here.

4) References - 2 (out of 39) are previous century cases, - perhaps you can remove them or replace with the modern ones? They do not fit for the respectable scientific Journal.

Author Response

Honored Professor,

Thank You very much indeed for the time spent reviewing my paper. We have been immensely delighted with the possibility of publication in Your Journal. All comments and recommendations on Your part were appreciated. We are convicted that considering the points discussed and alterations made, this manuscript will have its quality increased. All changes requested were made as follows:

  • Please, add the significance of the chosen mRNA, this should be described more in extent way (perhaps you can use some info from the Discussion and move it here);

Thank you for observing this significant lack of explanation, further information has been added about the decision to choose the miRNAs, and we have highlighted the added information.

  • please, put Ethical Committee permission info into the text. Additionally, please move info about the patients age, sex, anamnesis morbi et morborum main points, treatments etc. from Results Line134-140 to the Materials and methods Subject section. They fit here. Additionally, please, describe also the clear inclusion/exclusion criteria here about the patients and controls. Also, add the info were recurrent polyposis case included into the research group (separate then the primary and recurrent polyp group; now it is not understandable).

We agree with the necessity of further detailing the inclusion/exclusion criteria. All requested modifications have been made and are highlighted in Materials and Methods.

  • Conclusions. Please, make them more precise, - remove the 1st sentence, what is extra here.

Correction made

 References - 2 (out of 39) are previous century cases, - perhaps you can remove them or replace      with the modern ones? They do not fit for the respectable scientific Journal.

            We aknowledge the importance of adopting recent references in our article, however, the only two older articles we cited, are internationally recognized and standardized otorhinolaryngology scores. All specialty articles using these standard scores provide the same citation, the article where the scores were first described.

A new version of the manuscript, is being forwarded herein. Upon a careful reading of the manuscript, it became evident the improvements after reviewers’ analyses. If we have not reached the scope of the journal yet, we will be pleased to carry out any further modification(s). Thank You again and I remain sincerely yours.

Reviewer 2 Report

The study aimed to investigate the correlation between clinical parameters of chronic rhinosinusitis with nasal polyps (CRSwNP) and the expression of two microRNAs (miR-125b and miR-203a-3p). The expression levels of these two microRNAs in nasal polyps and normal nasal mucosa were determined using microarray analysis. miR-125b was found to be significantly overexpressed in nasal polyps and positively correlated with blood eosinophilia and nasal endoscopy score. MiR-203a-3p was found to be underexpressed in nasal polyps and significantly underexpressed in CRSwNP patients with environmental allergies.

1.      These two microRNAs (miR-125b and miR-203a-3p) have been studied to be related to chronic rhinosinusitis with nasal polyps. The increasing level of miR-125b in CRSwNP coincides with most literature, but the decreasing level of miR-203a-3p in CRSwNP does not coincide with the result in another paper.  Since miR-125b has been studied to be related to CRSwNP in the literature, what is the main contribution of this paper in studying the role of miR-125b? The conservation of this study for miR-203a-3p needs to be clarified and discussed.

Morawska-Kochman, Monika, et al. "Expression of Apoptosis-Related Biomarkers in Inflamed Nasal Sinus Epithelium of Patients with Chronic Rhinosinusitis with Nasal Polyps (CRSwNP)—Evaluation at mRNA and miRNA Levels." Biomedicines 10.6 (2022): 1400.

2.      Line 34. Provide the full name of EPOS20.

3.      Line 51. Typo error.

4.      Table 1. hasa-miR-203a-3p hsa-miR-203a-3p

5.      Line 110. table 1 →”Table 1.

6.      Lines 183-184. It is unclear.

7.      Lines 187-188. Should miR 123 be miR 125?

8.      Line 196, 209. micro-RNAs miRNAs

9.      Line 209, 211. microRNAs miRNAs. Sometimes you used microRNAs and sometimes you used miRNAs in the paper. It is not consistent

10.  Line 212. MiRNAs miRNAs

Author Response

Honored Professor,

Thank You very much indeed for the time spent reviewing my paper. We have been immensely delighted with the possibility of publication in Your Journal. All comments and recommendations on Your part were appreciated. We are convicted that considering the points discussed and alterations made, this manuscript will have its quality increased. All changes requested were made as follows:

  1. These two microRNAs (miR-125b and miR-203a-3p) have been studied to be related to chronic rhinosinusitis with nasal polyps. The increasing level of miR-125b in CRSwNP coincides with most literature, but the decreasing level of miR-203a-3p in CRSwNP does not coincide with the result in another paper.  Since miR-125b has been studied to be related to CRSwNP in the literature, what is the main contribution of this paper in studying the role of miR-125b? The conservation of this study for miR-203a-3p needs to be clarified and discussed. 

 Morawska-Kochman, Monika, et al. "Expression of Apoptosis-Related Biomarkers in Inflamed Nasal Sinus Epithelium of Patients with Chronic Rhinosinusitis with Nasal Polyps (CRSwNP)—Evaluation at mRNA and miRNA Levels." Biomedicines 10.6 (2022): 1400.

   We appreciate your keen observation, the explications provided in Introduction have been insufficient. We added more arguments explaining our choice of miRNAs- highlighted in Introduction, lines 68-71 and 77-83.

  Thank you very much for drawing attention over the article by Morawska-Kochman et al., which will bring complexity to our Discussion on miRNAs. We have mentioned and commented on this difference between the two articles- highlighted in Discussions, lines 296-300.

  1. Line 34. Provide the full name of EPOS20.- We added the full name, thank you for observing!
  2. Line 51. Typo error. – Corrected, thank you!
  3. Table 1. “hasa-miR-203a-3p” → ”hsa-miR-203a-3p”- Corrected, thank you!
  4. Line 110. “table 1” →”Table 1”.- Corrected, thank you!
  5. Lines 183-184. It is unclear. – This was an error, I apologize, and I have corrected.
  6. Lines 187-188. Should miR 123 be miR 125?- Corrected, thank you!
  7. Line 196, 209. “micro-RNAs” → ”miRNAs”- Corrected, thank you!
  8. Line 209, 211. “microRNAs” → ”miRNAs”. Sometimes you used microRNAs and sometimes you used miRNAs in the paper. It is not consistent

 You are absolutely right, thank you very much for drawing attention.

  1. Line 212. “MiRNAs” → ”miRNAs” - Corrected, thank you!

A new version of the manuscript, is being forwarded herein. Upon a careful reading of the manuscript, it became evident the improvements after reviewers’ analyses. If we have not reached the scope of the journal yet, we will be pleased to carry out any further modification(s). Thank You again and I remain sincerely yours.

Author Response

Honored Professor,

Thank You very much indeed for the time spent reviewing my paper. We have been immensely delighted with the possibility of publication in Your Journal. All comments and recommendations on Your part were appreciated. We are convicted that considering the points discussed and alterations made, this manuscript will have its quality increased. All changes requested were made as follows:

Line 51- please rewrite the sentence.

I apologize for this error, it was a writing error, and I have corrected it. Thank you!

Line 53- the sentence: ‘MiRNAs are a recently discovered family of small, non-coding RNAs responsible of inducing degradation or blocking translation of target mRNA, being considered the main regulatory mechanisms of gene expression[9, 10]’. Please cancel ‘recently’, because the discovery of miRNA is not recent, to date; in addition the ref. 9-10 could be changed with more recent works.

We appreciate your keen observation. We have corrected according to your suggestion. However, regarding references 9 and 10, we aknowledge the importance of adopting recent references in our article. However, the only two older articles we cited, are internationally recognized and standardized otorhinolaryngology scores. All specialty articles using these standard scores provide the same citation, the article where the scores were first described.

Line 56- sentence: ‘They play a significant role in cell division, differentiation and development, directly in modulating innate immune responses’,  is a limiting description of the regulating roles of miRNAs.

Thank you very much for this remark. We agree that the roles of miRNAs should not be diminished. We have added more details regarding their mechanism of action and roles- line 53-63.

Line 59- Please replaced the reference.

We have changed with a more complex reference. Thank you!

Discussion:

Line 187-189 correct the miRNA name. – Corrected, thank you!

Line 232-236 ….that patient symptom scores do not necessarily correlate with objective evaluations of disease severity…. Is true also for miRNAs? This sentence is questionable.

You are absolutely right, we have added further explications for this scentence- lines 255-258.

Line 238-239… By extrapolating the results, we could presume miR-125b as a candidate biomarker for severe CRSwNP… as reported previously (biomarker definition), this data is not sufficient to extrapolate the suggestion of miR-125b as biomarker, also miR-125b is over-expressed in many expression profile from several human diseases.

We agree, we have eliminated this scentence. Thank you!

Line 238-239… Please discuss the article by : Morawska-Kochman M, Śmieszek A, Marcinkowska K, Marycz KM, Nelke K, Zub K, Zatoński T, Bochnia M. Expression of Apoptosis-Related Biomarkers in Inflamed Nasal Sinus Epithelium of Patients with Chronic Rhinosinusitis with Nasal Polyps (CRSwNP)-Evaluation at mRNA and miRNA Levels. Biomedicines. 2022 Jun 13;10(6):1400. doi: 10.3390/biomedicines10061400.

Thank you very much for drawing attention over the article by Morawska-Kochman et al., which will bring complexity to our Discussion on miRNAs. We have mentioned and commented on this difference between the two articles- highlighted in Discussions, lines 297-301.

The entire work is based on the amplification of two miRNAs from polyps biopsy, and not only not was evaluated the full expression profile, useful to compare with other expression profile studies, but only one technical approach is not sufficient to assume significance. The full in vitro experiments are not performed, and all predictions are biased.

I suggest to provide a clear hypothesis by a vivid picture.

The current study is part of a PhD research thesis, and the technique selected and available for determining miRNA expression was real-time PCR.  The same technique for determining miRNA values from nasal polyps has been used in several similar publications, such as the following:

  1. Song, L., et al., Transcription Factor Specificity Protein 1 Regulates Inflammation and Fibrin Deposition in Nasal Polyps Via the Regulation of microRNA-125b and the Wnt/β-catenin Signaling Pathway. Inflammation, 2022. 45(3): p. 1118-1132.
  2. Yu, J., et al., Gene Expression Profiles of Circular RNAs and MicroRNAs in Chronic Rhinosinusitis With Nasal Polyps. Front Mol Biosci, 2021. 8: p. 643504.
  3. Liu R, Du J, Zhou J, Zhong B, Ba L, Zhang J, Liu Y, Liu S. Elevated microRNA-21 Is a Brake of Inflammation Involved in the Development of Nasal Polyps. Front Immunol. 2021 Apr 15;12:530488.
  4. Xia, G., et al., Differentially Expressed miRNA in Inflammatory Mucosa of Chronic Rhinosinusitis. J Nanosci Nanotechnol, 2015. 15(3): p. 2132-9.
  5. Zhang, X.H., et al., Overexpression of miR-125b, a novel regulator of innate immunity, in eosinophilic chronic rhinosinusitis with nasal polyps. Am J Respir Crit Care Med, 2012. 185(2): p. 140-51.

As far as in vitro experiments are concerned, the present study does not utilize cell cultures, all quantification of miRNA-125b and miRNA-203a-3p has been performed directly on patient tissue, comparing polypoid tissue with normal mucosa. We have provided statistically significant results and figures regarding the values we observed for the studied miRNAs. Moreover, for miRNA-125b we managed to demonstrate a statistically significant directly proportional correlation between objective clinical measurements of disease severity and values of miRNA in nasal polyps. Having these results in mind, and comparing them to similar findings in literature, we presumed correct to confirm miRNA-125b as a possible biomarker for CRSwNP, and propose miRNA-203a-3p as a valuable, but yet insufficiently investigated biomarker for this pathology.

A new version of the manuscript, is being forwarded herein. Upon a careful reading of the manuscript, it became evident the improvements after reviewers’ analyses. If we have not reached the scope of the journal yet, we will be pleased to carry out any further modification(s). Thank You again and I remain sincerely yours.

Round 2

Reviewer 2 Report

The authors have addressed my comments.

Reviewer 3 Report

Now after corrections the paper is accepted.